
# Conductance asymmetries in mesoscopic superconducting devices due to finite bias

André Melo[1,★], Chun-Xiao Liu[1,2], Piotr Rożek[1,2],
Tómas Örn Rosdahl[1] and Michael Wimmer[1,2]

**1** Kavli Institute of Nanoscience, Delft University of Technology,
Delft 2600 GA, The Netherlands
**2** QuTech, Delft University of Technology, Delft 2600 GA, The Netherlands

★ am@andremelo.org

## Abstract

Tunneling conductance spectroscopy in normal metal-superconductor junctions is an important tool for probing Andreev bound states in mesoscopic superconducting devices, such as Majorana nanowires. In an ideal superconducting device, the subgap conductance obeys specific symmetry relations, due to particle-hole symmetry and unitarity of the scattering matrix. However, experimental data often exhibits deviations from these symmetries or even their explicit breakdown. In this work, we identify a mechanism that leads to conductance asymmetries without quasiparticle poisoning. In particular, we investigate the effects of finite bias and include the voltage dependence in the tunnel barrier transparency, finding significant conductance asymmetries for realistic device parameters. It is important to identify the physical origin of conductance asymmetries: in contrast to other possible mechanisms such as quasiparticle poisoning, finite-bias effects are not detrimental to the performance of a topological qubit. To that end we identify features that can be used to experimentally determine whether finite-bias effects are the source of conductance asymmetries.



## 1  Introduction

Hybrid nanostructures combining spin-orbit coupled semiconducting nanowires and conventional superconductivity are a promising candidate to host Majorana bound states (MBS) [1–16]. Much of the ongoing experimental work on these devices relies on two-terminal tunnel spectroscopy in which the nanowires are coupled to a normal reservoir through an electrostatic tunnel barrier. In the tunneling limit the conductance through the normal-superconductor (NS) junction is proportional to the local density of states at the edge of the nanowire. This allows to measure local signatures of MBS such as a resonant zero-bias conductance peak [17–27]. Additionally, a three-terminal setup allows to probe nonlocal conductances, which can provide information about the bulk topology and the BCS charge of bound states [28–30].

A common theoretical framework for calculating the conductance in NS junctions is the scattering matrix ($S$) method under the linear response approximation [31]. In the presence of particle-hole symmetry and unitarity of the $S$ matrix, the linear response conductance $G$ obeys several symmetry relations at voltages below the superconducting gap $\Delta$. In two-terminal setups, for example, the conductance is symmetric about the zero bias voltage point, i.e., $G(V) = G(-V)$ for $|V| < \Delta/e$ [32, 33]. In three-terminal setups, it has recently been shown that the anti-symmetric components of the local and nonlocal conductance matrices are equal [30]. However, in experiments these symmetry relations are only observed approximately [34–39]. So far, possible mechanisms for the observed deviations that have been discussed in the literature always rely on coupling to a reservoir of quasiparticles, for example through dissipa tion due to a residual density of states in the parent superconductor or additional low-energy states [33, 40], or inelastic relaxation processes connecting subgap states to the above-gap continuum [41].

In this work we go beyond the linear response regime and study how finite-bias effects break conductance symmetry relations, without the need for quasiparticle poisoning. In particular, we consider the dependence of the tunnel barrier profile and transparency on the applied bias voltage in the normal lead [32, 42, 43]. In two-terminal setups, we find that a voltage-dependent tunnel barrier introduces asymmetry in both the width and height of subgap conductance peaks. Moreover, we study the conductance asymmetry as a function of system parameters, and show that it is enhanced by mirror asymmetric barrier shapes. We also identify general features that can be used to experimentally determine whether finite-bias effects are the main source of conductance asymmetry. Finally, we turn our attention to three-terminal setups and observe that finite-bias effects break conductance symmetries in accordance with recent experimental work [39].

## 2  Finite-bias conductance in a mesoscopic superconducting system

The formalism for computing the nonlinear conductance in a mesoscopic superconducting device has been derived in [32]. We give a concise summary here to point out the important

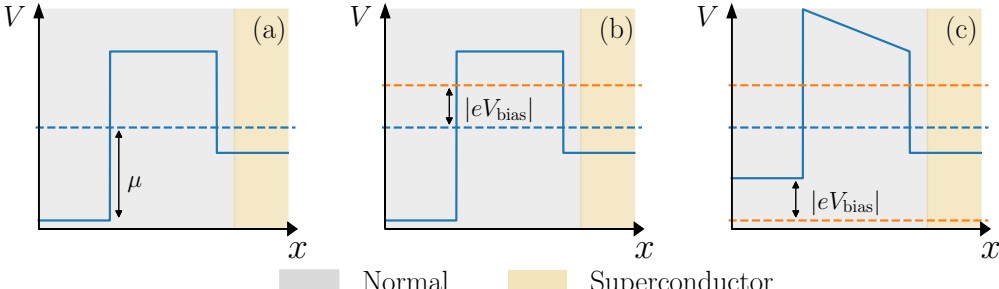

Figure 1: (a) Schematic band diagram of an NS tunnel junction at zero bias voltage. (b) Calculating conductance through the junction in the linear response limit. The voltage dependence of the scattering region is neglected and therefore the scattering matrix depends solely on the energy of incoming modes. (c) Finite-bias conductance includes changes in the electrostatic profile of the junction due to the applied voltage, e.g. a positive shift of the chemical potential near the normal lead, along with a linear voltage drop across the tunnel barrier. As a result the scattering matrix depends on both the energy of the incoming modes and the applied bias voltage.

aspects for our study.

Consider a scattering region attached to a normal lead and a superconducting lead shown schematically in Fig. 1(a). Using the Landauer-Buttiker formalism [31,44] we write the current in the normal lead as a sum of three contributions

$$I^{(\text{e})} = -\frac{e}{h} \int dE \, f(E + eV_{\text{bias}})[N(E, V_{\text{bias}}) - R_{ee}(E, V_{\text{bias}})], \tag{1}$$

$$I^{(\text{h})} = \frac{e}{h} \int dE \, f(E - eV_{\text{bias}}) R_{eh}(E, V_{\text{bias}})$$

$$= \frac{e}{h} \int dE [1 - f(E + eV_{\text{bias}})] R_{he}(E, V_{\text{bias}}), \tag{2}$$

$$I^{(\text{sc})} = \frac{e}{h} \int dE \, f(E) T_{es}(E, V_{\text{bias}}), \tag{3}$$

where $e = |e|$ and we have set the chemical potential of the superconductor to zero. $I^{(\text{e})}$ ($I^{(\text{h})}$) is the current carried by electrons (holes), $I^{(\text{sc})}$ the current originating from quasiparticles in the superconducting lead, and

$$f(E) = \frac{1}{1 + \exp\left(\frac{E - \mu}{k_B T}\right)}, \tag{4}$$

is the Fermi-Dirac distribution. $N$ is the number of electron modes in the normal lead, $R_{ee}$ the total electron reflection amplitude, $R_{eh}$ the total Andreev reflection amplitude and $T_{es}$ the transmission amplitude from the superconductor above-gap modes. In contrast with the conductance obtained in the linear response approximation, the finite-bias conductance takes into account changes in the profile of the tunnel barrier due to the applied bias voltage $V_{\text{bias}}$ (Fig. 1(c)). Therefore $R_{ee}$, $R_{eh}$ and $T_{es}$ depend not only on the energy of incoming particles $E$, but also on $V_{\text{bias}}$. Unitarity of the scattering matrix implies that

$$N(E, V_{\text{bias}}) = R_{ee}(E, V_{\text{bias}}) + R_{eh}(E, V_{\text{bias}}) + T_{es}(E, V_{\text{bias}}). \tag{5}$$

Hence

$$I^{(\text{sc})} = \frac{e}{h}\int dE\, f(E)[N(E,V_{\text{bias}}) - R_{ee}(E,V_{\text{bias}}) - R_{eh}(E,V_{\text{bias}})]$$

$$= \frac{e}{h}\int dE\, f(E)[N(E,V_{\text{bias}}) - R_{ee}(E,V_{\text{bias}})] - \frac{e}{h}\int dE[1 - f(E)]R_{he}(E,V_{\text{bias}})$$

$$= \frac{e}{h}\int dE\, f(E)[N(E,V_{\text{bias}}) - R_{ee}(E,V_{\text{bias}}) + R_{he}(E,V_{\text{bias}})]$$

$$- \frac{e}{h}\int dE\, R_{he}(E,V_{\text{bias}}). \tag{6}$$

The total current is then given by

$$I = \frac{e}{h}\int dE[f(E) - f(E + eV_{\text{bias}})][N(E,V_{\text{bias}}) - R_{ee}(E,V_{\text{bias}}) + R_{he}(E,V_{\text{bias}})]. \tag{7}$$

In the zero-temperature limit the conductance reduces to [32]

$$G = \frac{dI}{dV_{\text{bias}}} = \frac{e^2}{h}\left(N(-eV_{\text{bias}},V_{\text{bias}}) - R_{ee}(-eV_{\text{bias}},V_{\text{bias}}) + R_{he}(-eV_{\text{bias}},V_{\text{bias}})\right)$$

$$- \frac{e}{h}\int_0^{-eV_{\text{bias}}} dE\left[\frac{\partial R_{he}(E,V_{\text{bias}})}{\partial V} - \frac{\partial R_{ee}(E,V_{\text{bias}})}{\partial V}\right]. \tag{8}$$

Equation (8) is the most general form of finite-bias conductance. It does not assume any specific electrostatic profile of the junction and is also valid for multi-terminal setups. When the dependence of the NS junction on the applied bias voltage is ignored (Fig. 1(b)), that is $R_{ij}(E,V_{\text{bias}}) \to R_{ij}(E,0)$, Eq. (8) reduces to the well-known expression for NS conductance in the linear response limit

$$G_{\text{lin}}(V_{\text{bias}}) = \frac{2e^2}{h}\left(N - R_{ee}(-eV_{\text{bias}}) + R_{he}(-eV_{\text{bias}})\right), \tag{9}$$

which satisfies the symmetry relation $G(V_{\text{bias}}) = G(-V_{\text{bias}})$ at voltages below the superconducting gap [32, 33].

## 3 Finite-bias local conductance into a single Andreev bound state

To obtain a qualitative understanding of the influence of finite-bias effects, we first consider a toy model of an NS junction where the nanowire hosts a single Andreev bound state:

$$H = E_0 \begin{pmatrix} 1 & 0 \\ 0 & -1 \end{pmatrix}. \tag{10}$$

Below the superconducting gap, Eq. (8) reduces to

$$\frac{G}{2G_0} = R_{he}(-eV_{\text{bias}},V_{\text{bias}}) - \frac{1}{e}\int_0^{-eV_{\text{bias}}} dE\,\frac{\partial R_{he}(E,V_{\text{bias}})}{\partial V}, \tag{11}$$

where $G_0 = \frac{e^2}{h}$ is the conductance quantum. $R_{he}(E,V)$ can be obtained by taking the trace over the appropriate block of the scattering matrix, which we compute through the Mahaux-Weidenmüller formula

$$S = \mathbb{1} - 2\pi W^\dagger(E - H + \pi WW^\dagger)^{-1}W, \tag{12}$$

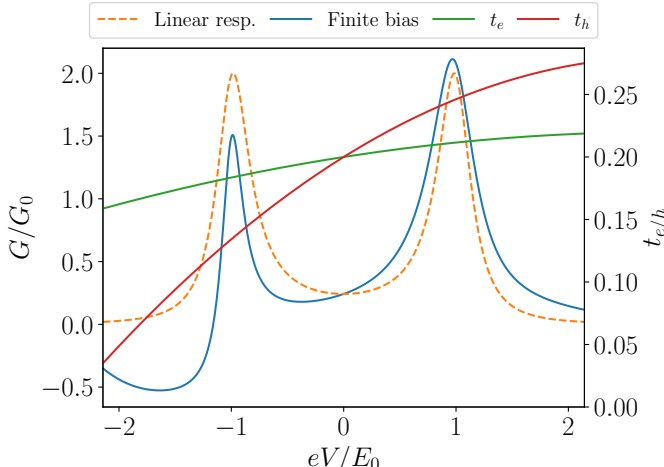

Figure 2: Two-terminal NS conductance into a single Andreev bound state with a scattering region sensitive to the applied bias voltage. In the linear response approximation the voltage dependence of the scattering region is neglected, resulting in particle-hole symmetric conductance profiles (orange dashed lines). When a voltage dependence is included in the electron/hole tunneling amplitude $t_{e/h}$ (red/green solid lines), the corresponding conductance profiles (blue solid lines) show different heights and widths at positive and negative bias voltage.

where

$$W = \begin{pmatrix} u t_e(E,V) & v^* t_h(E,V)^* \\ v t_e(E,V) & -u^* t_h(E,V)^* \end{pmatrix}, \tag{13}$$

parameterizes the coupling of the bound states to the lead modes. For notational convenience we drop the $E$ and $V$ dependencies of $t_{e/h}$ below, but their presence should be kept in mind.

We start by computing the first term in Eq. (11). The Andreev reflection amplitude is given by

$$16\pi^2 E_0^2 |uv t_e t_h|^2 \cdot \left\{ (E^2 - E_0^2)^2 + \pi^2 [2E E_0(|u|^2 - |v|^2)(|t_e|^4 - |t_h|^4) \right.$$
$$\left. -4E_0^2 |uv|^2 (|t_e|^2 - |t_h|^2)^2 + (E^2 + E_0^2)(|t_e|^4 + |t_h|^4)] \right\}^{-1}, \tag{14}$$

where we assume the junction is in the tunneling limit so that $t_{e/h} \ll E_0$ and we can safely discard terms of order higher than $\mathcal{O}(t_{e/h}^4)$. In the vicinity of $E = E_0$ we obtain the approximate expression

$$R_{he}(-eV_{\text{bias}}, V_{\text{bias}}) \approx \frac{4\pi^2 |t_e t_h uv|^2}{(-eV_{\text{bias}} - E_0)^2 + \pi^2 (|t_e u|^2 + |t_h v|^2)^2}. \tag{15}$$

Hence, the first term in Eq. (11) gives a Lorentzian conductance profile with a resonance at $|V| = E_0/e$. The height and full-width half maximums of the resonances are given by

$$\frac{G_{\text{max}}}{2G_0} = \frac{4|t_e t_h uv|^2}{(|t_e u|^2 + |t_h v|^2)^2}, \tag{16}$$

$$\text{FWHM} = \frac{2\pi}{e} \left( |t_e u|^2 + |t_h v|^2 \right). \tag{17}$$

The expressions for $V = -E_0/e$ can be readily obtained through the transformation $u \leftrightarrow v$. In the linear response regime we have $t_{e/h}(E, V_{\text{bias}}) = t_{e/h}(E, 0)$. Particle-hole symmetry gives

the constraint $t_e(E,0) = t_h(-E,0)$ and thus the subgap conductance is also particle-hole symmetric. However, when finite-bias effects are included, $t_{e/h}(\pm E_0, \pm E_0/e)$ are not constrained to be equal, resulting in particle-hole asymmetric conductance.

The contribution of the second term in Eq. (11) is (see App. A for the full calculation)

$$-\frac{2e}{h}\int_0^{-eV_{\text{bias}}} dE\, \frac{\partial R_{he}}{\partial V} = \left[ A \arctan\left(\frac{2(E-E_0)}{\text{FWHM}}\right) + B\frac{E-E_0}{\left(\frac{\text{FWHM}}{2}\right)^2 + (E-E_0)^2} \right]_0^{-eV_{\text{bias}}}, \quad (18)$$

where

$$A = -\frac{G_{\max}\cdot\text{FWHM}}{2G_0|t_e t_h|}\frac{\partial |t_e t_h|}{\partial V}\Bigg|_{\substack{E=E_0\\V=-E_0/e}} + \frac{\pi^2 G_{\max}}{2G_0 e^2\cdot\text{FWHM}}\frac{\partial\left(|ut_e|^2 + |vt_h|^2\right)^2}{\partial V}\Bigg|_{\substack{E=E_0\\V=-E_0/e}} \quad (19)$$

$$B = \frac{\pi^2 G_{\max}}{2G_0 e^2\cdot\text{FWHM}}\frac{\partial\left(|ut_e|^2 + |vt_h|^2\right)^2}{\partial V}\Bigg|_{\substack{E=E_0\\V=-E_0/e}}. \quad (20)$$

Both terms in Eq. (18) vary on the scale of FWHM and therefore do not change the width of the Lorentzian peaks in Eq. (15). However, they change the height of Eq. (16) by $\approx -\pi A/2 - B/E_0$.

To compute the conductance of the toy model, we choose $u = v = 1/\sqrt{2}$ and expand the tunnel rates about $E = V_{\text{bias}} = 0$ up to second order:

$$t_{e/h}(-eV_{\text{bias}}, V_{\text{bias}}) \approx t_{e,h}(0,0) + a_{e/h}V_{\text{bias}} + b_{e/h}V_{\text{bias}}^2. \quad (21)$$

The remaining parameters can be found in the accompanying code for the manuscript [45]. We show the resulting finite-bias conductance profile along with the corresponding linear response conductance in Fig. 2. In accordance with the analytical results in Eqs. (16) and (17), the finite-bias conductance peaks exhibit height and width asymmetry. Moreover, we observe that the finite-bias conductance has a region with negative values, which is due to the presence of the integral term. In contrast, the linear response conductance must always be positive.

## 4 Tight binding simulations

### 4.1 Finite-bias local conductance in a normal/superconductor geometry

To investigate finite-bias effects at a more realistic level, we consider a one-dimensional semiconductor-superconductor nanowire coupled to a normal lead. The Bogoliubov-de Gennes Hamiltonian for the NS junction can be written as

$$H = \left(\frac{p_x^2}{2m_{\text{eff}}} + \alpha p_x \sigma_y - \mu(x) + V(x, V_{\text{bias}})\right)\tau_z + V_Z\sigma_x + \Delta(x)\tau_x, \quad (22)$$

where $\sigma_i$ and $\tau_i$ are Pauli matrices acting in spin and Nambu space, $p_x = -i\hbar d/dx$, $m_{\text{eff}}$ the effective mass, $\mu$ the chemical potential, $V$ the onsite electrostatic potential, $\alpha$ the strength of Rashba spin-orbit interaction, $V_Z$ the Zeeman spin splitting, and $\Delta$ the superconducting gap. In particular, the chemical potential is a piecewise constant function of $x$ as

$$\mu(x) = \begin{cases} \mu_{\text{lead}}, & x < 0 \\ \mu_{\text{wire}}, & x > 0, \end{cases} \quad (23)$$

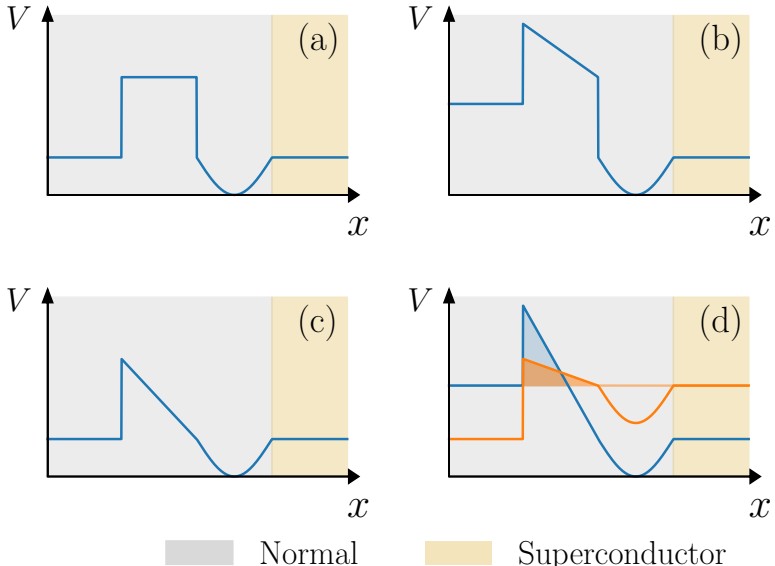

Figure 3: NS junctions with a bias-dependent tunnel barrier and a quantum dot potential. In the top figures we show a square barrier with at (a) zero bias voltage, and (b) negative bias voltage. In the bottom figures we show a triangular barrier at (c) zero bias voltage, and (d) at negative bias voltage (blue curve) and positive bias voltage (orange curve). The shaded regions indicate the effective barrier seen by an incoming electron at $E = -eV_{\text{bias}}$. When the barrier is triangular shaped, the effective barrier at positive voltage is smaller than at negative voltage, thus amplifying particle-hole asymmetry in conductance.

and the superconducting gap $\Delta(x)$ is finite only inside the nanowire.

The onsite potential has two terms $V(x, V_{\text{bias}}) = V_{\text{barrier}}(x, V_{\text{bias}}) + V_{\text{dot}}(x)$ illustrated in Fig. 3 (a)-(b). The first term corresponds to the electrostatic potential induced by the tunnel gate, which we model as a square barrier at equilibrium. A detailed calculation of the transport properties at finite bias requires a non-equilibrium approach [46]. However, in the tunneling regime the system is well approximated by the following phenomelogical model. When a bias is applied, the band bottom of the normal lead is shifted by $eV_{\text{bias}}$ and voltage drops linearly across the barrier [42, 47]:

$$V_{\text{barrier}}(x) = \begin{cases} -eV_{\text{bias}}, & x < 0 \\ eV_{\text{barrier}} - eV_{\text{bias}}(1 - \frac{x}{d}), & 0 \leq x < d \\ 0, & x > d. \end{cases} \tag{24}$$

Because the chemical potential of the lead also shifts by $-eV_{\text{bias}}$ when a voltage is applied, this potential keeps the charge density in the system constant. The second term is a smooth quantum dot potential [40]

$$V_{\text{dot}}(x) = \begin{cases} V_{\text{dot}} \cos\left(\frac{3(x-d)}{2L_{\text{dot}}}\right), & d < x < d + L_{\text{dot}} \\ 0, & \text{elsewhere,} \end{cases} \tag{25}$$

which induces a subgap Andreev bound state. In the following calculations and discussions we focus on how finite-bias effects cause particle-hole asymmetry for the Andreev bound state-induced resonance peaks at positive and negative bias voltages.

We apply the finite difference approximation to the continuum Hamiltonian (22) with a lattice constant of 1 nm, and numerically study the resulting tight-binding Hamiltonian using

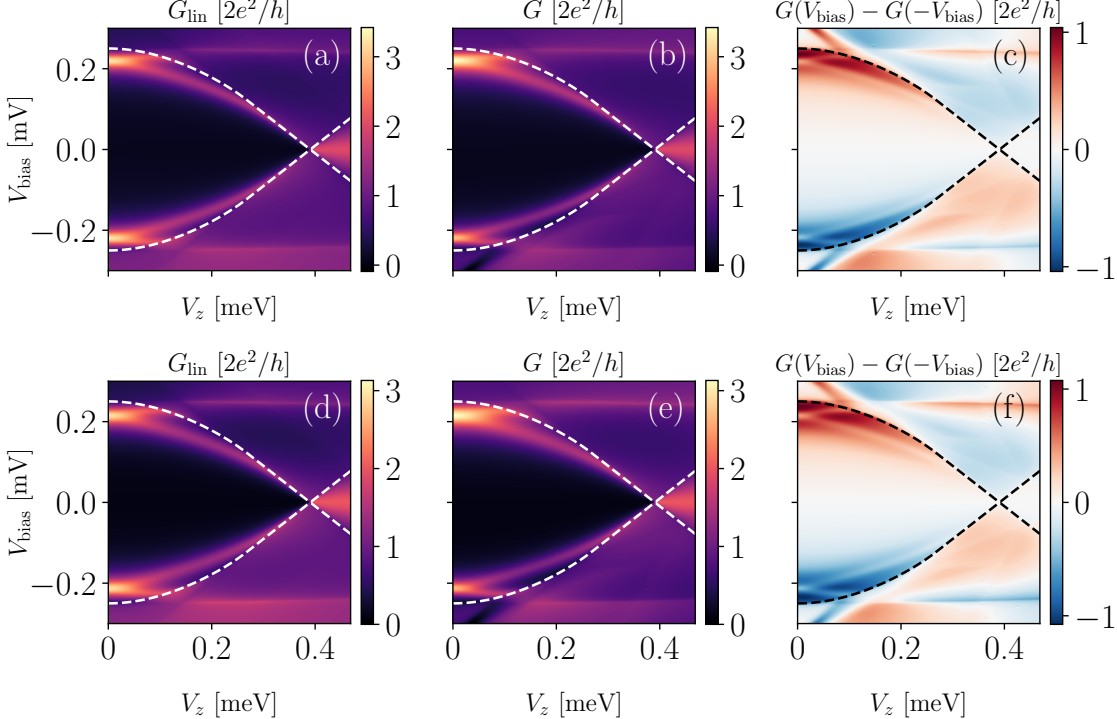

Figure 4: Two-terminal conductance as a function of Zeeman field and bias voltage in a proximitized nanowire with $\Delta = 0.25$ meV. The system in the top panels has a square barrier and in bottom panels a triangular barrier. The linear response conductance ((a) and (d)) is particle-hole symmetric below the induced gap (dashed lines). In contrast, the finite-bias conductance ((b) and (e)) shows significant particle-hole asymmetry below the gap which we plot explicitly in (c) and (f).

the Kwant software package [48]. Unless stated otherwise, the Hamiltonian parameters are $m_{\text{eff}} = 0.02m_e$, $\Delta = 0.25$ meV, $\alpha = 50$ meV nm, $V_{\text{dot}} = 2.2$ meV, $\mu_{\text{wire}} = 0.3$ meV, $\mu_{\text{lead}} = 0.55$ meV and the geometry parameters are $d = 80$ nm, $L_{\text{dot}} = 180$ nm. The source code and data used to produce the figures in this work are available in [45].

In Fig. 4(a) and (b) we show the linear response and finite-bias conductances as a function of bias voltage and Zeeman field strength. The Andreev bound state induces a resonance peak below the superconducting gap (white dashed line). Additionally, we plot the conductance asymmetry in Fig. 4(c). The conductance peaks display significant asymmetry in both their width and height. Furthermore, the magnitude of the asymmetry decreases as the peaks get closer to zero energy. This is a general feature of bias-induced asymmetry: states at higher energy have more asymmetry due to the larger effect on the electrostatic environments from the applied bias voltage. As a result, we expect that finite-bias effects will become more prominent as experiments begin to probe materials with higher superconducting gaps [49]. To illustrate this we consider a second nanowire with $\Delta = \mu_{\text{wire}} = 1$ meV, $\mu_{\text{lead}} = 3$ meV, $V_{\text{dot}} = 2.2$ meV and $L_{\text{dot}} = 50$ nm. Now the energy of the Andreev bound state is about four times larger than the previous case. The corresponding two-terminal conductance in Fig. 5(a)-(c) shows significantly more asymmetry than in the system of Fig. 4.

Besides the energy of the Andreev bound states, the transparency of the tunnel barrier also influences the conductance asymmetry. In Fig. 6 we plot the peak height asymmetries as a function of the barrier width and height of a square barrier for a system with $\Delta = 0.25$ meV and $V_Z = 0$. As the barrier height and width are increased, the relative importance of the

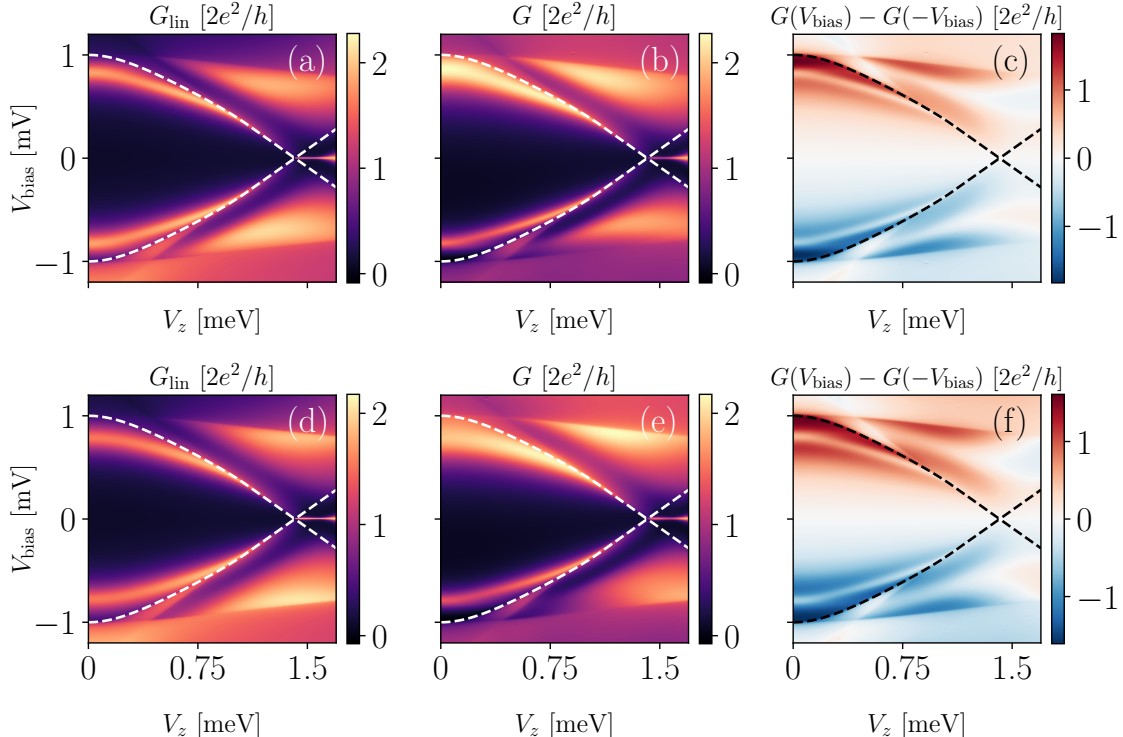

Figure 5: Two-terminal conductance as a function of Zeeman field and bias voltage in a proximitized nanowire with $\Delta = 1$ meV. The system in the top panels has a square barrier ($V_{\text{barrier}} = 0.8$ meV), and in bottom panels a triangular barrier ($V_{\text{barrier}} = 1.6$ meV). (a) and (d) show the linear response conductance, (b) and (e) the finite-bias conductance and (c) and (d) the asymmetry in finite-bias conductance.

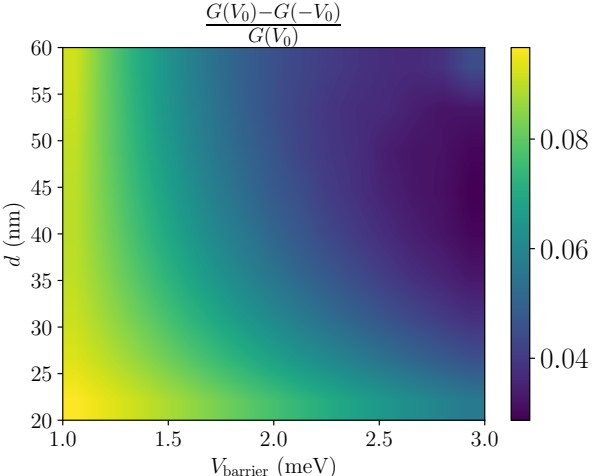

Figure 6: Normalized height asymmetry of conductance peaks in proximitized nanowire with $\Delta = 0.25$ meV and $V_Z = 0$ for varying barrier width $d$ and height $V_{\text{barrier}}$. The asymmetry vanishes as either the system is tuned deeper into the tunneling regime.

finite-bias modifications to the Hamiltonian decreases. Therefore the asymmetry decreases monotonically with both parameters, that is as the system is tuned deeper into the tunnel-

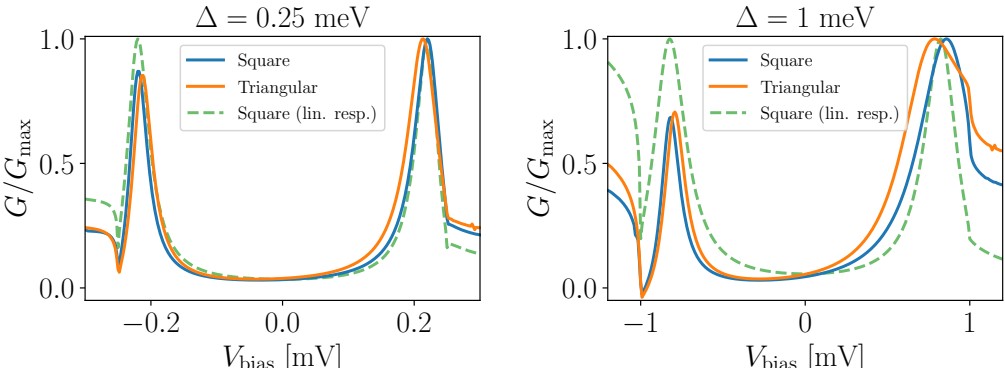

Figure 7: (a) Normalized finite-bias and linear-response conductances of systems with a square barrier and triangular barrier at $V_Z = 0$ and (a) $\Delta = 0.25$ meV and (b) $\Delta = 1$ meV (blue lines). Adding finite-bias effects breaks particle-hole symmetry of the linear response conductance (green dashed lines). A triangular tunnel barrier amplifies the width asymmetry of the peaks (though their height does not change significantly) because the effective barrier at positive voltages is smaller than at negative voltages.

ing regime. This behaviour is independent of the details of the barrier and thus is useful in determining finite-bias effects are the source of conductance asymmetry.

While the conductance asymmetry displays the general trends outlined above, its precise magnitude depends on the microscopic details of the scattering region, in particular in the barrier transmission probabilities at $\pm V_{\text{bias}}$. Within the WKB approximation the normal-state transmission probability is given by

$$T(E, V_{\text{bias}}) \propto \exp\left[-\frac{2}{h}\int_{\text{barrier}}\sqrt{2m(E - V(x, V_{\text{bias}}))}\right]. \tag{26}$$

The conductance through the barrier is therefore exponentially sensitive to the area of the barrier. In the case of a square barrier, these WKB areas are identical for $\pm V_{\text{bias}}$ due to the mirror symmetry of the barrier shape. However, if mirror symmetry in the barrier is broken, the effective WKB areas at negative and positive voltages become different, which further enhances the conductance asymmetry. As an example, we consider a system with a triangular barrier of height $V_{\text{barrier}} = 1.3$ meV, as illustrated in Fig. 3 (c)-(d)). In Fig. 4(c) we show the resulting conductance and see that it has larger particle-hole asymmetry than a system with a square barrier. This is more easily seen in Fig. 7(a)-(b) where we plot one-dimensional cuts of the conductance at $V_Z = 0$.

## 4.2 Finite-bias nonlocal conductance in a three-terminal geometry

In three-terminal devices with two normal leads coupled to a grounded superconductor the conductance is given by

$$G = \begin{pmatrix} G_{\text{LL}} & G_{\text{LR}} \\ G_{\text{RL}} & G_{\text{RR}} \end{pmatrix} = \begin{pmatrix} \frac{\partial I_L}{\partial V_L} & \frac{\partial I_L}{\partial V_R} \\ \frac{\partial I_R}{\partial V_L} & \frac{\partial I_R}{\partial V_R} \end{pmatrix}. \tag{27}$$

Because electrons can tunnel across the normal leads, the reflection matrix is not unitary below the gap. Hence the local conductance $G_{\text{LL}}$ is generally not particle-hole symmetric even in the

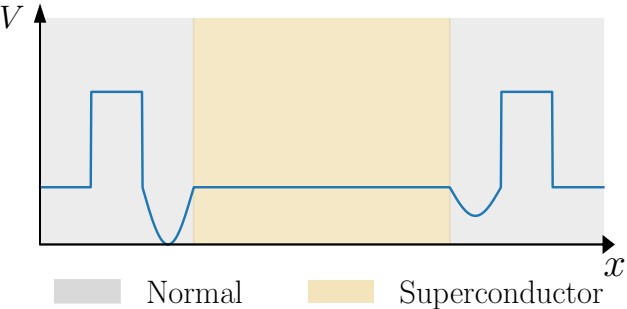

Figure 8: Schematic three-terminal superconducting device with bias dependent tunnel barriers and quantum dots.

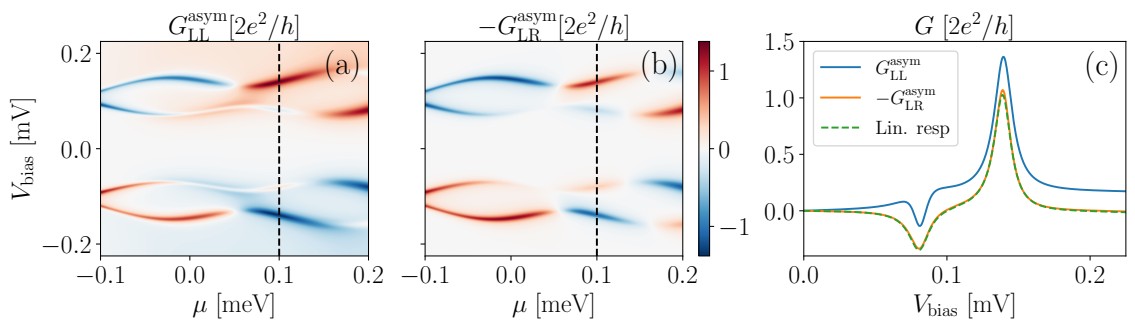

Figure 9: Anti-symmetric components of the (a) local conductance and (b) nonlocal conductance as a function of chemical potential and bias voltage. In panel (c) we show a one-dimensional cut of this data at fixed chemical potential (black dashed lines).

linear response limit.

However, a recent theoretical work showed that the anti-symmetric components of local an nonlocal conductances are related by [30]

$$G_{LL}^{\text{asym}} = -G_{LR}^{\text{asym}}, \tag{28}$$

where

$$G_{\alpha\beta}^{\text{asym}} = G_{\alpha\beta}(V_{\text{bias}}) - G_{\alpha\beta}(-V_{\text{bias}}). \tag{29}$$

Follow-up experimental data observed excellent agreement with this symmetry relation at low-bias voltages, but only qualitative agreement at high bias voltages [39]. Additionally $G_{LL}^{\text{asym}}$ and $G_{LR}^{\text{asym}}$ exhibited different behaviours near crossings of subgap states: while the crossings are avoided in $G_{LL}^{\text{asym}}$, they are unavoided in $G_{LR}^{\text{asym}}$.

To investigate whether finite-bias effects can explain these discrepancies, we consider a finite-length semiconductor-superconductor nanowire with length $L_{\text{sc}} = 300$ nm. On the right side of the device, we add another dot potential with $L_{\text{dot}}^{\text{left}} = L_{\text{dot}}^{\text{right}} = 350$ nm, and attach a second normal lead, as shown schematically in Fig. 8. When a bias is applied on the left (right) side, we drop the voltage across the left (right) barrier as specified in Eq.(24). Both the left and right potential wells host subgap Andreev bound states whose energies oscillate with chemical potential and display avoided crossings. However, due to the oscillatory nature of the wavefunction there are points in the parameter space in which the energy splitting of the states vanishes, similar to Majorana oscillations [50]. To avoid this and obtain spectra that

mimic those in [39] we break mirror symmetry and set $V_{\text{dot}}^{\text{left}} = 2 \cdot V_{\text{dot}}^{\text{right}} = 1$ meV. The remaining Hamiltonian parameters are the same as in Sec. 4.

In Fig. 9(a)-(b) we show the asymmetric components of the local and nonlocal conductances as a function of chemical potential and voltage, and in Fig. 9(c) we show a line cut at a fixed value of chemical potential. In accordance with the experimental results of [39], we observe $G_{LL}^{\text{asym}}$ and $G_{LR}^{\text{asym}}$ (orange and blue solid lines in Fig. 7(b)) show similar profiles qualitatively in general, but at the quantitative level, the deviation between them increases with the applied bias voltage, because the finite-bias effect is stronger at larger bias voltage as discussed in the previous sections. In contrast, the conductance components calculated under the linear response approximation are always equal to each other over the whole range of bias voltage (dashed line in Fig. 9(c)). However, our model does not capture the qualitative differences between $G_{LL}^{\text{asym}}$ and $G_{LR}^{\text{asym}}$ near avoided crossings. While this does not rule out finite-bias effects as the source of these discrepancies, it is also possible that they are caused by another physical mechanism.

## 5    Summary and discussion

In summary, we have shown that finite-bias effects in NS and NSN junctions can lead to significant deviations from linear response symmetries of the conductance matrix. In two-terminal NS junctions, the particle-hole symmetry between the conductance profiles at positive and negative voltages is broken, while for three-terminal NSN junctions, the equality between the asymmetric components of the local and nonlocal conductances no longer holds.

Although the exact values of the symmetry breaking depends on the details of the junction (e.g., the shape of the tunnel barrier and the magnitude of the superconducting gap), we find the asymmetry obeys two general qualitative trends. First, it decreases as the system is tuned deeper into the tunneling regime. Second, it grows with the applied bias voltage. As a result, finite-bias effects are more important in hybrid nanowires with a larger SC gap.

An important aspect about conductance asymmetries due to finite-bias effects is that they are not indicative of quasi-particle poisoning, unlike previously discussed mechanisms such as dissipation. Very recently, coupling of tunneling electrons to a phonon bath has also been predicted to give conductance asymmetries without quasiparticle poisoning [51]. Though originating from different physics, both mechanisms thus are not detrimental to Majorana qubits. Therefore, determining the source of conductance asymmetries is a helpful tool to predict qubit performance. The aforementioned trends allow to experimentally probe whether conductance asymmetries stem from finite-bias effects. As an example, if particle-hole symmetry of the conductance profiles in a two-terminal device is broken even when the bias voltage goes to zero [34, 36, 38], it is very likely that there are other mechanisms causing the symmetry breaking.

Finally, our treatment of the bias voltage dependence of the tunnel region is phenomenological. Future work could include computing finite-bias conductances with more realistic electrostatic potentials obtained by solving the self-consistent Schrödinger-Poisson equations [43, 52–54]. However, we expect that this will not change our qualitative findings.

## Acknowledgements

We are grateful to Anton Akhmerov for suggesting using the Mahaux-Weidenmüller formula to obtain analytical expressions for the conductance asymmetry, and thank J. Sau, D. Pikulin and T. Karzig for useful discussions.

**Author contributions**   M.W. formulated the project goal and oversaw the project with C.X.L. A.M. carried out the numerics with input from T.R. and P.R., and the analytical calculations with input from C.X.L. A.M. wrote the manuscript with input from the other authors.

**Funding information**   This work was supported by the Netherlands Organization for Scientific Research (NWO/OCW), as part of the Frontiers of Nanoscience program, an NWO VIDI grant 016.Vidi.189.180, an ERC Starting Grant STATOPINS 638760, and a subsidy for top consortia for knowledge and innovation (TKl toeslag) by the Dutch ministry of economic affairs and Microsoft research.

# A   Calculating the integral term of the conductance of a single Andreev bound state

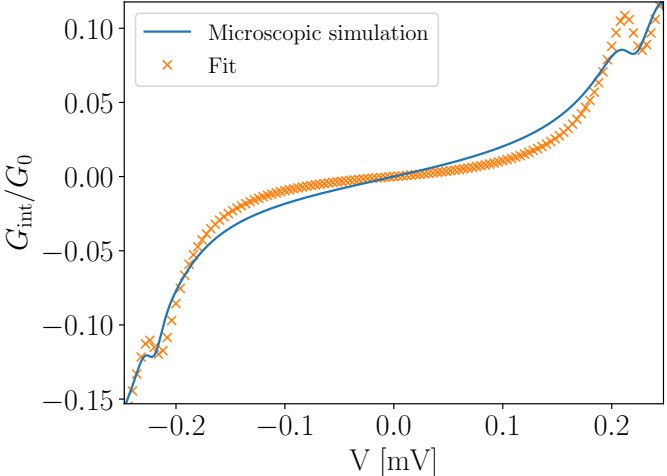

Figure 10: Blue line: integral term of the conductance for the system shown in Fig. 10(a) at $V_Z = 0$. Orange dots: fit to integral term expression obtained from two-level toy model.

To compute the integral term we start from the approximate form of Eq. (15). The derivative of $R_{he}$ with respect to $V$ is

$$\frac{\partial R_{he}(E,V)}{\partial V} = R_{he}(E,V) \left\{ \frac{2}{|t_e t_h|} \frac{\partial |t_e t_h|}{\partial V} - \frac{\pi^2 \frac{\partial}{\partial V} \left( |u t_e|^2 + |v t_h|^2 \right)^2}{(E-E_0)^2 + \pi^2 \left( |u t_e|^2 + |v t_h|^2 \right)^2} \right\}. \tag{30}$$

Because the integrand is sharply peaked at $E_0$ and we are interested in corrections near $eV_{\text{bias}} = E_0$ we approximate all derivatives of the tunneling rates as constant and evaluated at $E, eV_{\text{bias}} = E_0$. The contribution of the first term is then

$$-\frac{2}{e|t_e t_h|} \frac{\partial |t_e t_h|}{\partial V}\bigg|_{\substack{E=E_0 \\ V=-E_0/e}} \int_0^{-eV_{\text{bias}}} dE \, \frac{4\pi^2 |u v t_e t_h|^2}{(E-E_0)^2 + \pi^2 \left( |u t_e|^2 + |v t_h|^2 \right)^2} =$$

$$= -\frac{G_{\text{max}} \cdot \text{FWHM}}{2 G_0 |t_e t_h|} \frac{\partial |t_e t_h|}{\partial V}\bigg|_{\substack{E=E_0 \\ V=-E_0/e}} \left[ \arctan\left( \frac{2(E-E_0)}{e \cdot \text{FWHM}} \right) \right]_0^{-eV_{\text{bias}}}, \tag{31}$$

where we used the standard Lorentzian integral $\int dx \, \frac{a}{(x-x_0)^2+b^2} = \frac{a}{b}\arctan\left(\frac{x-x_0}{b}\right)$. The second term gives

$$
\begin{aligned}
&= \frac{\pi^2}{e}\frac{\partial}{\partial V}\left(|ut_e|^2+|vt_h|^2\right)^2\bigg|_{\substack{E=E_0\\V=-E_0/e}}\int_0^{-eV_{\text{bias}}}dE\,\frac{4\pi^2|uvt_et_h|^2}{\left((E-E_0)^2+\pi^2\left(|ut_e|^2+|vt_h|^2\right)^2\right)^2} = \\
&= \frac{2\pi|uvt_et_h|^2}{e\left(|ut_e|^2+|vt_h|^2\right)^3}\frac{\partial\left(|ut_e|^2+|vt_h|^2\right)^2}{\partial V}\bigg|_{\substack{E=E_0\\V=-E_0/e}} \\
&\quad\times\left[\frac{\pi\left(|ut_e|^2+|vt_h|^2\right)(E-E_0)}{\left(\pi\left(|ut_e|^2+|vt_h|^2\right)\right)^2+(E-E_0)^2}+\arctan\left(\frac{2(E-E_0)}{e\cdot\text{FWHM}}\right)\right]_0^{-eV_{\text{bias}}} = \\
&= \frac{\pi^2 G_{\max}}{2G_0e^2\cdot\text{FWHM}}\frac{\partial\left(|ut_e|^2+|vt_h|^2\right)^2}{\partial V}\bigg|_{\substack{E=E_0\\V=-E_0/e}} \\
&\quad\times\left[\frac{\frac{e\cdot\text{FWHM}}{2}(E-E_0)}{\left(\frac{e\cdot\text{FWHM}}{2}\right)^2+(E-E_0)^2}+\arctan\left(\frac{2(E-E_0)}{e\cdot\text{FWHM}}\right)\right]_0^{-eV_{\text{bias}}}.
\end{aligned}
\tag{32}
$$

Where we made use of the standard integral $\int dx \, \frac{a}{((x-x_0)^2+b^2)^2} = \frac{a}{2b^3}\left\{\frac{b(x-x_0)}{b^2+(x-x_0)^2}+\arctan\left(\frac{x-x_0}{b}\right)\right\}$. To test how well this expression works, we compute the integral term of the system shown in Fig. 4(a) at $V_Z = 0$ meV and fit it to

$$
G_{\text{int}} = \begin{cases} \left[A\arctan\left(\frac{2(E-E_0)}{\text{FWHM}}\right)+B\frac{E-E_0}{\left(\frac{\text{FWHM}}{2}\right)^2+(E-E_0)^2}\right]_0^{-eV_{\text{bias}}}, & V_{\text{bias}} < 0 \\ \left[C\arctan\left(\frac{2(E+E_0)}{\text{FWHM}}\right)+D\frac{E+E_0}{\left(\frac{\text{FWHM}}{2}\right)^2+(E+E_0)^2}\right]_0^{-eV_{\text{bias}}}, & V_{\text{bias}} > 0 \end{cases},
\tag{33}
$$

where $A$, $B$, $C$, $D$ are free parameters and $E_0$ are FHWM are measured from the conductance profile. The resulting fit is shown in Fig. 10.

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
