# Peer review of "Conductance asymmetries in mesoscopic superconducting devices due to finite bias"

_SciPost Physics, doi:SciPost Phys. 10, 037 (2021)_

## Round 1 · Referee Report · Anonymous (Referee 1) · 2020-10-7

Strengths

  • The paper is very clear and concise
  • It discusses an important and likely mechanism in current Majorana nanowire experiments
  • It combines numerical and analytical results
  • It offers sufficient motivation and context for the study
  • It discusses alternative mechanisms not included in their theory

Weaknesses

  • It is somewhat simplistic (it sticks to the simplest possible models and descriptions). This weakness is only relative, since it is probably the reason for the clarity of the work.
  • The results are not really "groundbreaking". They are merely useful and instructive.

Report

The paper is a very clear and concise discussion of the qualitative asymmetries in subgap NS and NSN conductance resulting from the dependence of barriers on bias voltages. The theory and models are sound and clearly highlight a likely and important effect in current experiments. The context and motivation are sufficient. I recommend publication in SciPost, as I consider the work satisfies the second acceptance criterion "Present a breakthrough on a previously-identified and long-standing research stumbling block" (although perhaps "breakthrough" and "long standing" may be overstatements)

Requested changes

Some caveats apply to the discussions and it would benefit the manuscript to spell them out a little bit more clearly

  • If I understand correctly, the whole discussion requires mu_{Lead} > Delta. Otherwise a -eV_{bias} < -mu_{Lead} would deplete the lead, and yield zero subgap conductance, unlike for the opposite bias. Hence, mu_lead<Delta would have asymmetric subgap conductance already in the linear response regime.

  • The spatially linear model assumed for the barrier profile under a bias voltage is an approximation. This should be stated more clearly when it is first introduced. Admitedtly, at the end of the paper this fact is mentioned, with a promise to discuss Schroedinger-Poisson solutions in future works. However, I think that is also somewhat misleading, as a rigorous treatment of the effect of a finite bias would require a non-equilibrium (Keldysh) formalism, not simply an equilibrium Schroedinger-Poisson approach. In any case, I agree that the phenomenological linear model probably does a perfectly fine job in the tunneling regime, and does not distract the reader with non-essential complexity.

  • Around Eq. (26) it is mentioned that barrier asymmetry can enhance conductance asymmetry. As it is phrased it can give the impression that barrier asymmetry alone could produce conductance asymmetries also in the linear regime. Careful reading and examination of Fig 4 show that this is not the case, but some more precise wording can help. Also, the mention of WKB transmission T can be confusing, as up to that point only NS reflections Ree and Reh were introduced. Maybe clarify that this is a normal-state transmission, and polish that part a bit more.

  • A little bit below Eq. (26), there is a typo in V_{Dot}^{Left} = 2 * V_{Dot}^{Left} = 1 meV

  • validity: high
  • significance: good
  • originality: high
  • clarity: top
  • formatting: perfect
  • grammar: perfect

Author:  André Melo  on 2021-01-28  [id 1183]

(in reply to Report 1 on 2020-10-07)

We thank the referee for the time and effort in reviewing our manuscript. Below we address the technical questions and remarks posed in the "Requested changes" section of the report.

1.The discussion in the paper is independent of the chemical potential in the lead, contrary to the referee's statement. As the referee points out setting $\mu_{lead} > \Delta$ and $-eV_{bias} < - \mu_{Lead}$ results in zero subgap conductance due to depletion of the electron sector in the lead. However, at the opposite bias voltages $-eV_{bias} > \mu_{Lead}$ it is instead the hole sector that is depleted, and hence Andreev reflection is also not possible. This also leads to zero subgap conductance and therefore the subgap conductance is still particle-hole symmetric. In fact, the original arguments in Ref. [32, 33] only depends on the existence of a scattering matrix, and does not put any restrictions on the chemical potential.

  1. We thank the referee for this remark. We have added a sentence in section 4 that clarifies this:

    A detailed calculation of the transport properties at finite bias requires a non-equilibrium approach [46].

  2. We have added some clarifying remarks to this paragraph.

  3. We thank the referee for noticing the typo, which we have fixed in the updated version.

---

## Round 1 · Referee Report · Anonymous (Referee 3) · 2020-10-12

Strengths

(1) The paper is concise, very well written, and clear.
(2) The paper discusses a topic that is timely and relevant
(3) The problem (asymmetry in the width and height of sub-gap tunneling conductance peaks in NS junctions for positive and negative bias voltages) has been defined clearly and a clear solution (dependence of the tunnel barrier profile and transparency on the sign of the applied bias voltage for finite-bias conductance) has been given.
(4) The presented solution is intuitive and supporting analytical and numerical evidence are presented.
(5) Although the referee does not see a groundbreaking discovery or theoretical/experimental breakthrough, the paper does present a new solution (dependence of the tunnel barrier on sign of the finite voltage) to an existing problem of sufficient interest (experimental observation of particle-hole symmetry breaking in two and three terminal subgap conductance in NS and NSN junctions)

Weaknesses

1) No obvious weakness.

Report

In this paper the authors discuss a timely problem, namely, asymmetry in the width and height of subgap conductance peaks in NS junctions for negative and positive finite bias voltage as well as related observations in NSN junctions. These questions are important for the ongoing experimental efforts to realize Majorana fermions in SM-SC heterostructures but will also have more general relevance for finite-bias tunneling conductance experiments through NS and NSN junctions. The authors present an intuitively clear explanation of this problem in terms of dependence of the tunneling barrier profile on the sign and magnitude of the finite bias voltage and provides supporting analytical and numerical evidence. I find the paper to be very well written and on an important topic that is very relevant and timely. I find no obvious weakness of this work except for the fact that it feels somewhat incremental. But having been associated with numerous papers in this field, I am also aware that papers of this kind, despite being important, almost always feel incremental. I believe the journal acceptance criteria have been met and the paper should be accepted for publication.

---

## Round 1 · Referee Report · Anonymous (Referee 2) · 2020-10-12

Strengths

  1. The manuscript proposes a mechanism for particle-hole asymmetry observed in tunneling into superconducting nanowires that might avoid the use of quasiparticle poisoning. This is potentially important for using tunneling transport to characterize topological nanowires for qubit applications.

  2. The proposed mechanism applies quite naturally to the semiconductor nanowire platform for topological superconductivity since the tunnel barriers involve small potential.

  3. The manuscript is quite clear and self-contained and is thus easy to follow even for beginners in the field.

  4. The manuscript provides both an analytic approach based on Eq. 14 as well as numerical results for a simple model for a topological nanowire.

Weaknesses

  1. It is somewhat unclear if the mechanism presented in the manucsript is large enough to account for the particle-hole aysmmetries measured in existing experiments. This is presumably left to future work.

  2. While the text below Eq. 26 and the caption of Fig 7 emphasize shape dependence of the particle-hole asymmetry, the effect seems to be of order 10% and mostly in the shape of the peak rather than the height.

  3. It is not clear that this mechanism is very relevant to transport in scanning tunneling microscopes where the barriers are likely higher and narrower.

Report

As listed in the strengths,the manuscript titled "Conductance asymmetries in mesoscopic superconducting devices due to finite bias" studies the effect of bias voltage on the barrier of transport measurements in topological superconducting nanowires.

I think the present manuscript is an important advance in the field because
it calls attention to this likely relevant aspect of transport in these
devices. Despite it's simplicity, this mechanism and its role in particle-hole
asymmetry seems to have been overlooked in the literature. It should be noted
that transport in these devices is particularly important as being one of
the only probes of topological superconductivity that we have experiments
with relatively clear interpretations (though still not being definitive).
Particle-hole asymmetry is an important quantity to study because it
is an indication of quasiparticle poisoning unless mechanisms such as the
one discussed in this manuscript are relevant. Quasiparticle-poisoning
is expected to be the main limiting factor in the reliability of topological
qubits constructed from topological superconductors. At any rate, one can
hope to improve our understanding of transport in topological nanowires.

For these reasons I recommend the manuscript for publication in Scipost
after the requested changes are made.

Requested changes

  1. The text around Eq. 26 that discusses Fig 7 as well as the caption of Fig 7 should be altered with a clarification of the type "while the shape of the barrier seems to modify the profile of the conductance peak, the peak height doesn't appear to be significantly altered by the barrier shape in our calculation."

  2. Please move this sentence "Besides the energy of the Andreev bound states, the geometrical shape of the tunnel barrier also plays an important role in the conductance asymmetry"
    on page 10 to where Fig 7 is discussed (likely the next paragraph). The present location of this sentence is not appropriate because the shape of the barrier is not discussed below this sentence or even near it.

  • validity: top
  • significance: high
  • originality: high
  • clarity: top
  • formatting: excellent
  • grammar: excellent

Author:  André Melo  on 2021-01-28  [id 1184]

(in reply to Report 2 on 2020-10-12)

We thank the referee for the time and effort in reviewing our manuscript. Below we address the technical questions and remarks posed in the "Requested changes" section of the report.

  1. Because the triangular barrier nevertheless amplifies the width asymmetry, we have added the following sentence:

    A triangular tunnel barrier amplifies the width asymmetry of the peaks (though their height does not change significantly) because the effective barrier at positive voltages is smaller than at negative voltages.

  2. We thank the referee for pointing out this innacuracy. We have rephrased the sentence to

    Besides the energy of the Andreev bound states, the transparency of the tunnel barrier also influences the conductance asymmetry.

---

## Round 2 · Author Response

Minor changes to address the referee reports and a correction in the analytical calculations of section 3. See "List of Changes" for more details.

---

## Round 2 · List of Changes

• Corrected the $W$ matrix (eq. 13) to ensure it has particle-hole symmetry. While this change leads to a different expression for the intermediate result in eq. 14, all other results remain valid.
  • Added clarification and fixes mentioned in replies to referee reports.

---

## Editorial Decision

published